# LLMs Struggle to Balance Reasoning and World Knowledge in Causal Narrative Understanding

**Khurram Yamin**
kyamin@andrew.cmu.edu
Carnegie Mellon

**Shantanu Gupta**
shantang@andrew.cmu.edu
Carnegie Mellon

**Gaurav Ghosal**
gghosal@andrew.cmu.edu
Carnegie Mellon

**Zachary Lipton**
zlipton@andrew.cmu.edu
Carnegie Mellon

**Bryan Wilder**
bwilder@andrew.cmu.edu
Carnegie Mellon

## Abstract

The ability to robustly identify causal relationships is essential for autonomous decision-making and adaptation to novel scenarios. However, accurately inferring causal structure requires integrating both world knowledge and abstract logical reasoning. In this work, we investigate the interaction between these two capabilities through the representative task of causal reasoning over narratives. Through controlled synthetic, semi-synthetic and real-world experiments, we find that state-of-the-art large language models (LLMs) often rely on superficial heuristics—for example, inferring causality from event order or recalling memorized world knowledge without attending to context. Furthermore, we show that simple reformulations of the task can elicit more robust reasoning behavior. Our evaluation spans a range of causal structures, from linear chains to complex graphs involving colliders and forks. These findings uncover systematic patterns in how LLMs perform causal reasoning and lay the groundwork for developing methods that better align LLM behavior with principled causal inference.

## 1 Introduction

Many successful applications of LLMs to causality leverage the ability of LLMs to absorb and summarize large amounts of world knowledge from large-scale unsupervised data. However, more ambitious roles for LLMs could require stepping beyond knowledge from pretraining and moving towards reasoning about causal structure in context, not merely recalling associations. In this work, we explore whether such contextual causal reasoning capabilities arises naturally from large-scale pretraining.

Robust causal reasoning is particularly challenging as it relies on a combination of knowledge and reasoning capabilities. On the one hand, causal reasoning relies on deductive or mathematical skills to correctly apply axioms (e.g., Pearl's do-calculus), making inferences from a graphical structure describing cause-effect relationships. However, unlike mathematical reasoning benchmarks (Cobbe et al., 2021) – which draw from a relatively constrained set of problem solving strategies and results – arriving at correct causal inferences often requires leveraging domain-specific knowledge about the events or variables involved to instantiate such a graph. These two abilities must be balanced; models must go beyond blindly retrieving memorized associations and knowledge to identify the correct relationships under atypical or counter-intuitive settings.

Prior works have primarily studied reasoning and world-knowledge of LLMs separately. For example, benchmarks on mathematical reasoning or coding typically study these capabilities in isolation – with minimal external world knowledge needed to solve problems. On the other hand, benchmarks for knowledge intensive tasks can generally be solved by simple retrieval of memorized knowledge. Thus, the interplay of knowledge retrieval and reasoning (and potential conflicts between them) remains understudied.

In this work, we study the interplay between *reasoning* and using the *right amount of world knowledge* through causal reasoning over textual narratives. In order to characterize LLMs' capabilities across the full range of interactions between these two characteristics, we need the ability to separately vary the difficulty of an instance along both dimensions. Accordingly, we construct a new set of tasks based on textual narratives generated from synthetic, semi-synthetic, and real-world causal relationships. Each instance starts from a true causal graph structure on a set of nodes $V_1...V_N$, from which we generate a narrative consistent with the true graph. Then, we present the LLM with only the narrative and ask it: (1) determine whether $V_i$ causes $V_j$ (directly or indirectly) for some pair $i, j$; and (2) given node identities $(V_1, \ldots, V_N)$, reconstruct a causal graph faithful to the narrative.

We then systematically control task difficulty along two axes. Along the *world knowledge conflict* axis, we manipulate how much the narrative diverges from memorized or "common-sense" causal knowledge (e.g., applications to atypical settings not commonly seen in pre-training). This tests the LLMs ability to reason using the actual context of the story as opposed to its memorized knowledge. Along the *graph reasoning complexity* axis, we vary the number of nodes in the underlying causal graph and the structure of the graph itself (e.g. simple chains versus graphs with complex structures including both forks and colliders). This tests the LLMs ability to extend its reasoning beyond simple scenarios to more complex situations.

Together, these design choices allow us to characterize LLMs' performance across the full spectrum of both dimensions of task difficulty. We find that gaps in performance across this spectrum are well-described by two distinctive failure modes related to interference between reasoning and world knowledge in causal inference. Firstly, we show that LLMs are influenced heavily by a prior that causes are likely to appear before effects in a narrative. We observe that when the narrative is constructed in the reverse topological order of the causal chain (i.e., the edge $V_i \rightarrow V_{i+1}$ is narrated *before* $V_{i-1} \rightarrow V_i$), the performance of the LLM suffers as it often assigns the cause to an earlier event and the effect to a later event in the narrative. Secondly, we show that LLMs use their parametric causal knowledge (i.e., if an event typically causes another event) as a shortcut to answer causal questions. Thus, when the cause-and-effect pairs implied by the narrative conflict with the parametric knowledge, the LLM often ignores the specifics of the narrative and defaults to its parametric knowledge. Neither prompting with Chain of Thought (CoT) (Wei et al., 2022) nor In-Context Learning alleviates these failures.

However, LLMs are much less impacted by variation in the reasoning difficulty of the task when the prompting scheme explicitly isolates reasoning and world knowledge. First, we find that asking the LLM to extract the entire causal graph implied by the narrative results in a high degree of success at correctly ordering individual events, largely avoiding both failure modes described above. However, these benefits dissipate if the model is prompted to use the extracted graph alongside the narrative. Second, LLMs exhibit only slight performance degradation when reasoning over narratives that display more complex graph structures than chains, for example forks or colliders. Third, while LLMs often struggle with longer narratives containing more events, this failure is also substantially mitigated by asking the LLM to just extract a graph. All together, our results paint a more nuanced picture of LLMs' causal capabilities than simple success or failure and suggest that future development should focus on isolating and then composing LLMs' strengths at reasoning and world knowledge in order to avoid conflicts between them.

## 2 RELATED WORKS

**Causal Reasoning in Large Language Models**  Jin et al. (2023) develop a benchmark for testing causal reasoning in LLMs given causal graphs, finding that language models can struggle with the task. However, the queries examined in Jin et al. (2023) require probability calculations, potentially conflating causal reasoning and arithmetic failures. Tan et al. (2022) shows the capability of a neural network trained on news data to label causal structures in individual sentences. Joshi et al. (2024b)

chronicles failure modes in textual, but non-narrative form data (e.g. text formulaically written as Event 1 Causes Event 2 Causes Event 3 Causes Event 4). Our paper expands upon such a line of work by testing the LLM's abilities in both real and synthetic texts that much more closely resemble those seen in everyday life. Another contrasting work, Jin et al. (2024), uses only statistical language indicating event correlations as input.

(Gordon et al., 2012; Joshi et al., 2024a; Ho et al., 2023; Zhang et al., 2023; Wang et al., 2023; Ashwani et al., 2024) study causal reasoning ability as it relates to inferring causal relations based on "common sense". In such common-sense based settings, it is straightforward for models to simply rely on memorized knowledge from pretraining and achieve good performance, without leveraging any more general causal reasoning capabilities. Our work seeks to disentangle this general causal reasoning ability by specifically testing cases where causal relationships may contradict common-sense knowledge. This serves as a more robust measurement of the causal reasoning capabilities in unfamiliar and atypical scenarios. Empirically, we show that models struggle significantly in adapting to unfamiliar causal relations.

Another important distinction of our work is the focus on longer-form narratives. Existing works such as (Gordon et al., 2012; Zečević et al., 2023; Ho et al., 2023; Frohberg & Binder, 2022; Li et al., 2023; Gao et al., 2023) primarily examine short-form questions about a single causal relationship. On the other hand, our work examines longer and more complex sequences of events. Moreover, in contrast to domain-specific question banks such as Intuitive Physics studied in (Zečević et al., 2023), our narratives examine a more diverse range of topics (as illustrated by the sample narratives presented). As a result, our dataset provides a more realistic and diverse examination of LLM causal reasoning capabilities than prior works. As such, our work is unique in that we are the first paper to analyze non-common sense based causal reasoning in narratives that use everyday language.

**Causal Story Generation**   Kıcıman et al. (2024) shows that LLMs have strong abilities to generate causal texts. Ammanabrolu et al. (2020) introduces soft causal relations—causal constraints that match what readers expect—and uses commonsense inferences to bridge high-level plot points, resulting in more coherent narratives that align with everyday causal expectations. Tian et al. (2021) contributes by employing counterfactual knowledge to generate hyperboles, making story generation more realistic. Li et al. (2022) shows that asking a model to explain a cause or effect by generating new text conflates language generation with prediction; instead, their approach asks the model to simply indicate the sentence number representing the cause or effect, leading to stories that better respect causal relations. For our synthetic text generation, we focus on creating narratives that are extremely explicit and simple. In contrast to Ammanabrolu et al. (2020) and Li et al. (2022)'s approach of bridging events using commonsense, our narrative scheme already embeds explicit causal language between events that are causally related so that no inference or common-sense reasoning is required from the reader to reason about causality. Furthermore, our experiments often intentionally contradict common-sense parametric knowledge to check the model's ability to solely rely on the self-contained narrative. Similarly, regarding Tian et al. (2021), we opted to avoid abstract language structures that might confuse even human readers.

## 3 Experiments with Synthetic Data

### 3.1 Setting

**Synthetic Narrative Generation**   In our synthetic experiments, we use three leading LLMs: OpenAI's GPT-4o (OpenAI et al., 2024), Anthropic's Claude 3.5 Sonnet (Anthropic, 2024), and the open source LLama 3.1 8b (Grattafiori et al., 2024). While we focus on GPT-4o in the main text, results from other models are in the Appendix. The purpose of our synthetic setup is to carefully control the conditions under which the LLMs are tested. In terms of the general setup of our fully synthetic experiments, we first use the LLM to generate events (which are real world phenomena like *rain* or *plants growing*). Then these events are linked together into a chain graph $G$ that acts as the causal ground truth (eg *rain* $\rightarrow$ *plants growing*). The LLM is given $G$ and asked to create a narrative that stays faithful to the causal relationships in $G$. These narratives are checked by researchers to ensure consistency with their base causal graphs. More specifically, when constructing the dataset, we asked researchers (3 non-author graduate students who were blind to the true underlying graph) to reconstruct the causal chains given just the narratives, and over 98 percent of the time (out of 150

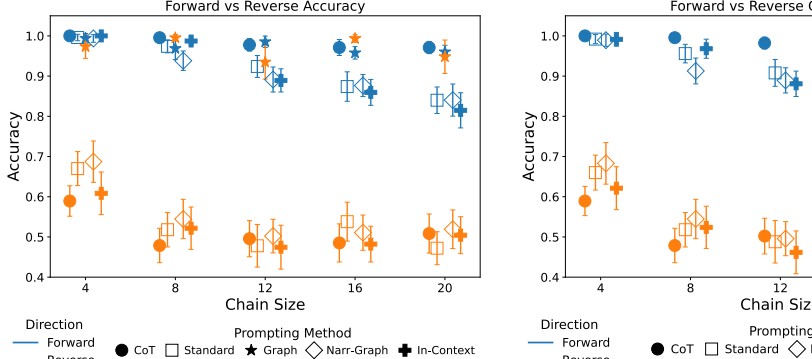

Figure 1: GPT-4o Test of the LLM's ability to reason on narratives written in the Forward and Reverse topological orientations. Chain size is the number of nodes in ground truth $G$. The "Graph" prompting method uses only the extracted graph $G'$ to reason, "Narr-Graph" uses both the narrative and extracted graph, and "Standard, CoT, In-Context" all use only the narrative. Accuracy measures LLM answer agreement with $G$ (we test every possible ordered pair of variables and check whether the extracted graph correctly implies the existence and direction of the corresponding causal edge when compared to the ground truth $G$), and consistency measures agreement with $G'$. The points in the graph are represented with a slight horizontal stagger around the relevant chain sizes (4,8,12 etc) for ease of visual understanding. We show a 95% CI.

random samples), the humans were able to find the unique correct causal ordering (Appendix F). Roughly 2500 narrative samples were generated. To ensure a variety of events go into the narratives, we generate 100 to 1000 distinct events at a time and randomly pick the small number needed for narrative construction (all narratives in supplementary files and select narratives in Appendix).

Providing only the narrative as input (and not $G$), we then ask the LLM to find $G'$, the predicted underlying causal structure expressed by the narrative. In other words, the LLM is asked to output a causal graph that it thinks embodies the relationships in the narrative. Next, a series of causal questions is created by randomly sampling 10 tuples of events from $G$ and asking the LLM whether an event in the tuple causes the other based on the narrative and/or $G'$. All results are taken and aggregated over 5 random seeds, with the CI being taken after aggregation.

**Prompting Strategies** We evaluate five prompting styles for causal reasoning where the names in italics represent those used in the legends of figures: **Standard QA Prompting** (*Standard*), where the model is simply asked to identify the causal relation between two narrative events; **Chain-of-Thought** (*CoT*), which instructs the model to articulate step-by-step reasoning before answering; **In-Context Learning** (*In-Context*), which precedes the query with illustrative question–answer examples; **Explicit Causal Graph Extraction** (*Graph*), which asks the model to generate an entire causal graph $G'$ over all events and assesses whether the ordering of the target pair is correct; **Narrative-Augmented Graph Extraction** (*Narr-Graph*), which first elicits $G'$ and then supplies both $G'$ and the original narrative for joint reasoning about the causal pair. Exact prompts are in Appendix A.

## 3.2 IMPACT OF EVENT ORDERING

Our experiments show that LLMs rely on the ordering in which the events are verbalized in a narrative when determining causal relationships. To investigate this, we started with randomly generated events that were used to make a ground truth graph $G$. During the creation of the narrative, we specified that the LLM either places the events in (1) the order that matches the topological causal ordering of the graph (e.g., if event $A$ (indirectly or directly) causes $B$, then event $A$ is mentioned before $B$ in the narrative), or (2) a way that runs opposite to the causal ordering (event $B$ would be mentioned before $A$ in the narrative even though $A$ (directly or indirectly) causes $B$). We refer to these as the *Forward* and *Reverse* topological ordering, respectively. As an example, the following is a

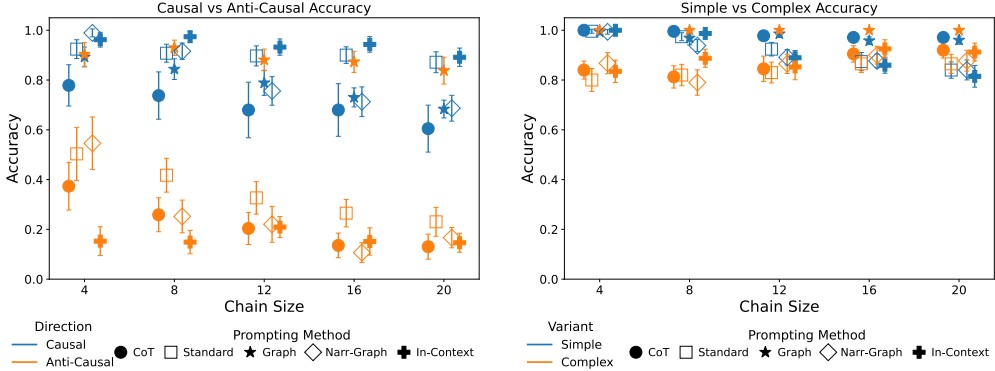

Figure 2: (Left) GPT-4o test of the LLM's ability to reason on narratives that agree with parametric knowledge (Causal) and disagree with parametric knowledge (Anti-Causal). (Right) GPT-4o test of the LLM's ability to reason on narratives generated from Complex graphs as opposed to Simple chain graphs. Label descriptions for both images match those of Figure 1 and 95 % CI is shown.

GPT-4o generated *Reverse* topological narrative for the causal chain: *Art exhibition→ Wine tasting → Charity fundraiser*:

> The *charity fundraiser* was made possible because of the successful *wine tasting event* that attracted numerous generous patrons. The *wine tasting* was organized as a result of the *art exhibition* drawing in a sophisticated audience interested in cultural experiences.

Each edge in the narrative is verbalized in the opposite order to its place in the causal chain. All narratives can be found in the linked code.

**LLMs Rely on Event Ordering Across Prompting Strategies** As shown in Figure 1 (left), in the *Forward* direction, standard QA, CoT, and In-Context prompts all perform very well. This is in contrast to the *Reverse* orientation when we look at the performance of the standard QA, COT, and In-Context prompts. From this plot, we can see that naive COT and In-Context prompting do not seem to significantly boost accuracy under our conditions. Perhaps more interestingly, we find that the way the LLM answers questions using the narratives is not always consistent with the causal graph $G'$ that the LLM builds when asked to predict the underlying graph structure (see consistency plot in right side of Figure 1, where consistency measures agreement between the answers of the LLM and $G'$ whereby we see what percent of answer implied by $G'$ are also implied by the LLM). In the *Reverse* orientation, answers given by the extracted causal graph G' and the previously discussed prompting strategies seem to differ greatly. Additionally, the trend of those prompting strategies on the consistency plot for the *Forward* orientation narratives (comparing performance to $G'$) mirrors their trend on the accuracy plot which compares performance to ground truth $G$ (left side).

**Explicit Causal Graph Extraction Avoids Shortcuts** This led us to test the accuracy of only using the extracted graph $G'$ to answer causal questions (Figure 1, "Graph" Method) . In this case, once $G'$ is extracted by the LLM, it is not given to the LLM again to answer questions (but rather used directly with a graph traversal). We found that this strategy did significantly better in the *Reverse* direction than the other prompting strategies ($\sim$ 50 % better). Surprisingly, using $G'$ in the *Reverse* direction narratives to answer causal questions did as well as using $G'$ in the *Forward* direction narratives. Next, we tried prompting using the narrative and $G'$ (the LLM is given $G'$ in this case in the prompt). This technique could be thought of as a type of CoT prompting strategy. However, in the *Reverse* direction narratives, the increase in accuracy achieved by only using $G'$ completely dissipates. We conjecture that the process of building the extracted Causal Graph $G'$ forces the LLM to engage in long term reasoning instead of using the simple shortcut, but when the narrative is again provided - the LLM defaults back to the shortcut.

### 3.3 Impact of Parametric Knowledge (In)consistency

**Experimental Setup**   We also find that LLMs tend to rely on parametric knowledge when it is present, and can fail when narratives are inconsistent with the LLM's parametric knowledge. To test this, we elicit the LLM's pre-existing parametric knowledge when generating the event chains. We prompt the LLM to pick a series of events such that each event has some relation to the subsequent event – either the event is *Causal* to the next event (e.g., disease causes shorter lives) or the event is *Anti-Causal* (e.g., disease causes longer lives). For example, we might know that node 1 is *Anti-Causal* to node 2 from parametric knowledge. Thus, when we make the causal ground truth graph $1 \rightarrow 3 \rightarrow 2$ (this disagrees with parametric knowledge), create a narrative from it, and then ask the LLM if node 1 causes 2 based on the narrative: it should say yes based on the narrative even though that disagrees with its parametric knowledge. After the ground truth graph is created, we generate the narrative in the *Forward* topological orientation to avoid confounding failure modes. The full process (along with illustration) explaining how the parametric and causal graphs are created is in Appendix A.2. As a textual example, assume that we know a parametric anti-causal link exists from *stressful job* to *increased happiness*, and from *lack of sleep* to *improved cognitive function*. We can then construct the causal chain *Stressful Job → Lack of Sleep → Increased Happiness → Improved Cognitive Function*. From this causal chain, we create the narrative:

> The constant demands of a *stressful job* led to her experiencing chronic *lack of sleep*. Surprisingly, she found that the *lack of sleep* heightened her sense of euphoria, making her unusually cheerful at work. *Increased happiness* from this unexpected cheerfulness seemed to improve her *cognitive function*.

If the LLM is asked if a stressful job leads to increased happiness, the parametric knowledge shortcut indicates the answer should be no – however, the shortcut fails as the narrative indicates that a (indirect) causal link does exist.

**Models Exploit Parametric Knowledge**   We find that, in synthetic experiments, the LLM finds the correct causal relation generally only when that relation agrees with its parametric knowledge. This is exemplified in the plot in Figure 2 (left) where we see good performance on narratives that agree with parametric knowledge (*Causal* parametric knowledge) and poor performance on narratives that disagree with parametric knowledge (*Anti-Causal* parametric knowledge). We also notice an interesting phenomenon for the *Anti-Causal* parametric case where using just the extracted graph provides massive improvements over any prompting strategy that involves using the narrative to directly answer questions. This strategy is comparable in performance to when the parametric knowledge is *Causal*. It seems that the narrative may only serve to distract the LLM when parametric knowledge disagrees with the narrative.

### 3.4 Impact of Narrative Complexity

In the previous sections, we identified two shortcuts which models exploit in causal reasoning tasks. Here, we test the influence of narrative complexity on these failure modes. We examine two measures of complexity: (a) the narrative length and (b) the presence of complex graph structures.

**Narrative Length**   In conditions where the LLM exhibits failure modes (*Reverse* and *Anti-Causal* orientations), the performance also tends to decay as the size of the narrative and the number of events in the narrative increases. As we can see in Figures 1 and 2 (Left), it seems that the longer the narrative is, the more the LLM relies on shortcuts instead of performing reasoning. However, the extracted graph $G'$ can often maintain a consistently high level of accuracy across narrative sizes even for cases when a failure mode would normally be exhibited.

**Causal Graph Complexity**   As the bulk of our work has focused on detecting the simplest failure modes possible, we studied narratives with an underlying chain graph structure. However, the presence of more complex causal structures in the narrative could exacerbate the existing failure modes or trigger novel failures. To study this, we create causal graphs utilizing two common causal structures: *Forks* (one node has a causal relationship to multiple other nodes) and *Colliders* (multiple nodes have a causal relationship to the same node). We generate narratives (the complete algorithm is described in Appendix A.3) such that each underlying causal graph contains at least one of these

structures, and may randomly contain multiple such structures based on the size of the narrative. An example is shown in Figure 3.

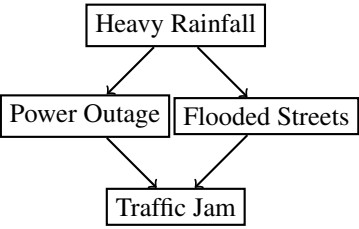

**Narrative:** The *heavy rainfall* not only caused a *power outage* in several neighborhoods but also led to *flooded streets*. The aftermath of the *power outage* (disabling traffic lights) and the *flooded roads* (blocking street access) caused a *traffic jam*.

Figure 3: Causal graph with story showing a fork (first sentence) and a collider (second sentence).

As can be seen in Figure 2 (right side), we find that while the LLM generally performs worse at reasoning about the complex narratives than simple narratives (with underlying chain graphs), the gap is very starkly less than can be seen in the other failure modes. This finding can be supported by (Dettki et al., 2025) which finds that GPT-4o reasons similarly to humans on a single sentence that describes one collider relation. Our work extends their work by using a long-form narrative based on a causal graph with potentially multiple colliders and forks instead of only one collider.

## 4 EXPERIMENTS WITH REAL WORLD CAUSAL GRAPHS

In this section, we extend our analysis to narratives involving real-world causal graphs from *CauseNet* (Heindorf et al., 2020), a large-scale knowledge graph of (claimed) causal relationships between real-world concepts. We perform experiments using the *GPT-4o* (OpenAI et al., 2024) and Llama-3.1 8B models for our experiments. We concentrate our analysis on the same factors (positional biases and parametric knowledge consistency) as explored in the semi-synthetic settings.

The *CauseNet* dataset can be represented as a collection of $D$ tuples $\{(C_i, E_i, \mathbf{S}_i\}_{i=1}^{D}$, where $C_i$ denotes the cause (e.g., fatigue), $E_i$ denotes the effect (e.g., accidents), and $\mathbf{S}_i$ is a set of sentences (extracted from Wikipedia and ClueWeb12 (Callan, 2012)) that entail a causal relationship from $C_i$ to $E_i$. We retrieve causal chain graphs $V_1 \rightarrow V_2 \rightarrow \ldots \rightarrow V_N$ of various lengths, where each causal relation $V_i \rightarrow V_{i+1}$ is from *CauseNet* and verbalize these chains as narratives in the following ways:

**Semi-synthetic narratives.** In this setting, we use real causal graphs from *CauseNet* but synthetically verbalize them via the LLM. In particular, we prompt the LLM to generate sentences for each edge $(V_i \rightarrow V_{i+1})$ in the causal graph, while ensuring the sensibility of the entire narrative. For eammple, the following is a narrative for the chain *fatigue → accidents → injury*:

> *Fatigue* can cloud judgment and slow reaction times, leading to an increase in *accidents* on the road. As a result, these *accidents* often lead to serious *injury* for those involved, highlighting the dangerous consequences of driving while fatigued.

**Real-world narratives.** For the real-world narratives, the sentence for each edge is chosen from the *CauseNet* dataset. To ensure that the narrative as a whole remains coherent, we prompt the LLM to ensure that the sentences for every pair of adjacent edges logically follow each other. For example, the following is the narrative for the causal chain *fatigue → accidents → injury*:

> Workers work long hours in mines and factories where *fatigue* and a lack of concentration can easily cause *accidents*. These *accidents* are the leading cause of *injury* in this country for people ages 1-34.

Additional examples of semi-synthetic and real-world narratives are presented in Appendix C.1 (the entire set of narratives used for our experiments is available in the linked code).

**Prompting Strategies** For simplicity, we limit the prompting techniques used to (see Appendix C.2 for the prompt templates): **Standard QA Prompting**, **Chain-of-Thought** and **Explicit Causal Graph Extraction**. We evaluate the accuracy for each pair of nodes $(V_i, V_j)$ for the three prompting strategies on the semi-synthetic and real-world narratives.

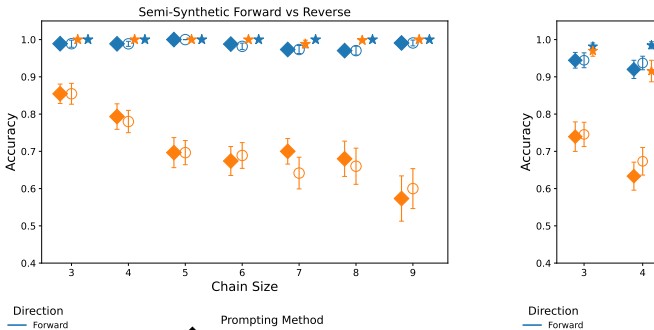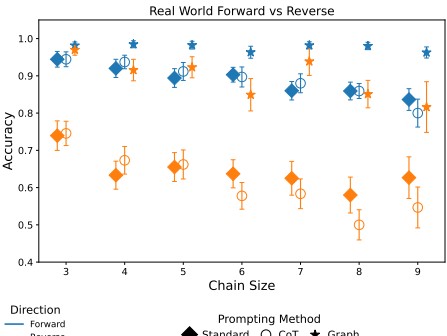

Figure 4: The accuracy of various prompting strategies (error bars denote 95% CIs). We observe that the accuracy is lower in the reverse direction (and tends to decay as the chains get longer).

## 4.1 IMPACT OF EVENT ORDERING AND CHAIN LENGTH

As described in the previous section, we verbalize each causal chain graph $V_1 \rightarrow V_2 \rightarrow \ldots \rightarrow V_N$ from *CauseNet* into a narrative in the forward and reverse topological order. In both the semi-synthetic (Fig. 4 left) and real-world narratives (Fig. 4 right), the *Forward Graph* strategy performs the best, with its accuracy remaining stable even as the chain length increases. We observe that *Forward Standard and CoT* outperforms *Reverse Standard and CoT*, with the *Reverse* accuracy declining substantially as the chain size gets large. We also see that in this regime, extracting the causal graph makes inference in the *Reverse* orientation competitive with inference in *Forward*.

## 4.2 EFFECT OF PARAMETRIC KNOWLEDGE CONSISTENCY

**Experiment Setup** Next, we analyze the extent to which the LLM relies on its parametric knowledge to answer causal reasoning queries as opposed to the causal structure expressed in the narrative. For every pair of nodes $(V_i, V_j)$ in the chain graphs, we elicit the parametric knowledge of the LLM by asking the LLM whether a causal effect between the two nodes would be atypical (see Appendix B.2 for the exact prompts utilized). Through these prompts, we identify cause and effect chains which contradict the model's parametric knowledge. For example, in a chain graph from our dataset, there is a path from *streambank erosion* to *higher prices*, but this contradicts the LLM's parametric knowledge since this causal effect may not typically exist in the real-world. In total, we find that roughly 5 percent of the relations in CauseNet violate the LLM's pretraining knowledge. We sampled narratives from CauseNet until we got 100 (of chain sizes between 3 and 9) narratives that contain relations that violate the LLM's pre-training knowledge and 100 that are consistent. These narratives are constructed in the *Forward* topological ordering to avoid confounding failure modes.

**LLM Performance Suffers on Atypical Causal Relations** We evaluate the three prompting strategies separately on the subsets of cause-and-effect pairs that are in agreement and in conflict with the parametric knowledge (see Table 1). We observe that when there is no conflict (i.e., the parametric knowledge agrees with the causality expressed in the narrative), the accuracies with and without CoT are greater than 90%. However, when the parametric knowledge conflicts with the narrative's causality, the accuracy is significantly lower, even with CoT. This suggests that when asked to reason about cause and effect in a narrative, the LLM seems to rely heavily on its parametric knowledge and is unable to grasp the specific causal chains expressed in the narrative itself (despite the causal chains as a whole being realistic).

**Explicit Causal Graph Extraction Avoids Shortcuts** Interestingly, when using extracted graph for performing causal reasoning, the performance is very high, both with and without conflicts. This is likely because when asked to extract the graph from the narrative, the LLM pays more attention to the entire narrative as opposed to when directly queried on a cause-and-effect pair (where the LLM defaults to its parametric knowledge). These results show that even when the LLM constructs a reasonably good causal chain graph, the LLM does not leverage this graph when queried directly

|  | Standard | CoT | Graph |
|---|---|---|---|
| **Semi-synthetic** | | | |
| Without Conflict | 99.8 | 99.6 | 99.9 |
| With Conflict | 67.2 | 73.1 | 98.7 |
| **Real-world** | | | |
| Without Conflict | 90.9 | 89.2 | 97.9 |
| With Conflict | 52.1 | 57.6 | 93.2 |

Table 1: The average accuracy across different narratives with the three prompting strategies partitioned by whether the cause-effect pairs conflict with the LLM's parametric knowledge (we omit the 95% CIs as they are smaller than 0.3).

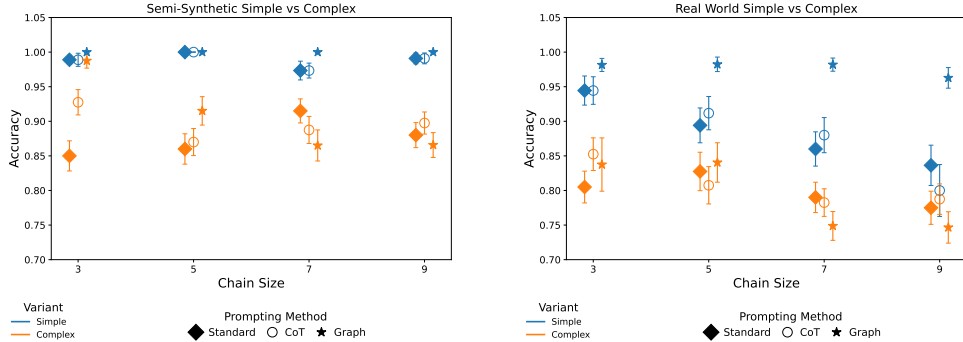

Figure 5: GPT-4o accuracy on narratives generated from Complex graphs as opposed to Simple chain graphs for semi-synthetic narratives (left) and real-world narratives (right). 95 % CI is shown.

about the causal effects in the narrative (even with CoT), further highlighting the advantage of extracting the causal graph directly.

### 4.3 Narrative Complexity

We can see from Figure 4 that LLM performance degrades with narrative length, especially when a failure mode is present. We furthermore experimented with complex narratives with causal graphs containing forks and colliders (full graph and narrative creation algorithm in Appendix B.3). We can see in Figure 5, that in both the semi-synthetic and real-world settings that complex narratives (with colliders and forks) perform worse than simple narratives that have a causal chain graph as the ground truth. This gap ,while clear and noticeable, isn't as stark as failure from parametric knowledge conflict (Table 1) or topological ordering (Figure 4). We do furthermore note that this is one area where extracting an explicit causal graph does not seem to significantly improve performance.

### 5 Discussion

Our work takes initial strides towards examining the success and failure of LLMs to reason causally on narratives that express causal events. We focus on two questions of key importance in causality: (1) Does one event cause another? (2) Can the LLM extract the causal graph from the narrative. We find three significant failure modes of LLM reasoning by conducting experiments in carefully controlled synthetic, semi-synthetic and real-world settings: Firstly, we find that LLMs rely heavily on **topological ordering**, performing well when the ordering of events in the narratives matches that of the ordering of the underlying causal graph. Secondly, we find that LLMs rely on their **parametric knowledge** as a shortcut to infer causal relations. Finally, we examine the role of **causal structure complexity**, finding that LLM accuracy degrades as the narrative length increases. Furthermore, LLMs perform slightly worse on reasoning when narratives contain structures such as colliders and forks. Beyond these failure modes, we show that more reliable causal reasoning can be elicited by prompting the LLM to explicitly identify the causal graph. One limitation of our work is that

there are other forms of causal reasoning that we did not test for in the narratives. This motivates many potential directions for future work. For example, it could be interesting to ask the LLM to reason about counterfactual cases. Our analysis also has implications for algorithmic interventions to improve causal reasoning. The failure modes we identify in this paper could inform the design of targeted synthetic tasks to use in finetuning for improved causal reasoning. Additionally, our findings on the benefits of extracting a causal graph can inform prompt engineering efforts to elicit reliable causal reasoning from language models. We believe investigating both directions represents an exciting direction for future work.

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

APPENDIX

# A  SYNTHETIC DATA EXPERIMENTS

## A.1  SELECTED SYNTHETIC PROMPTS

We use an LLM to generate the events $E$. From the events, we create a ground truth causal graph $G$ which is used to structure and inform the narrative sequence and causality. $N$ is the corresponding narrative created by the LLM from $G$. To evaluate the LLM's performance, we extract a causal graph, $G'$, from the narrative $N$ as produced by the LLM, and compare it with the ground truth causal graph $G$. In this context, $n$ refers to the number of events to generate, while $A$ and $B$ represent pairs of events queried for causal relationships. The task then becomes assessing whether event $A$ causes event $B$. All prompts, data processing steps, and results are included in the attached code. Furthermore all results are taken and aggregated over 5 random seeds, with the CI being taken after aggregation.

### A.1.1  TOPOLOGICAL EXPERIMENT - GENERATING RANDOM EVENTS ($E$)

"generate $n$ random distinct events"

### A.1.2  PARAMETRIC EXPERIMENT - GENERATING A PAIR OF CAUSAL EVENTS ($E$)

"generate a pair of events that cause each other. generate an event that causes another event, for example Cancer $\rightarrow$ Death or Obesity $\rightarrow$ Bad Heart Health. Make sure the event generated is not already in $E$ "
This is repeated as many times as is necessary

### A.1.3  PARAMETRIC EXPERIMENT - GENERATING A PAIR OF ANTI-CAUSAL EVENTS ($E$)

"generate a pair of events that are anticausal (an event causing the opposite of the normal effect), for example the first event could be cancer and the second event could be a longer life because in reality, cancer causes a shorter life. Make sure the events generated are not already in $E$."
This is repeated as many times as is necessary

### A.1.4  FORWARD TOPOLOGICAL NARRATIVE ($N$)

"Output a short narrative (use one sentence) that expresses the causal link [E1 $\rightarrow$ E2]. By causal link, we mean that the sentence should convey that E1 directly caused E2. In other words, it should be clear from the narrative that E2 would not have happened had E1 not happened. Ensure that the words [E1, E2] are present in the new sentence and E1 appears before E2. Only output the new sentence."
Repeat for all causal/anti-causal links

### A.1.5  REVERSE TOPOLOGICAL NARRATIVE ($N$)

"Output a short narrative (use one sentence) that expresses the causal link [E1 $\rightarrow$ E2]. By causal link, we mean that the sentence should convey that E1 directly caused E2. In other words, it should be clear from the narrative that E2 would not have happened had E1 not happened. Ensure that the words [E1, E2] are present in the new sentence and E2 appears before E1. Only output the new sentence."
Repeat for all causal/anti-causal links

### A.1.6  STANDARD PROMPT

"Use this narrative $N$ as context. Did $A$ cause $B$? Output your answer with $<answer>Yes/No</answer>$. The cause can be direct or indirect."

### A.1.7  IN-CONTEXT PROMPT

"Use this narrative $N$ as context. Did $A$ cause $B$? Output your answer with $<answer>$ Yes/No $</answer>$. The cause can be direct or indirect. An example narrative would be: Rains leads to plants growing. This then causes increased oxygen in the atmosphere. A potential question would be:

does rain cause increased oxygen in the atmosphere? The answer would be Yes. Another example narrative would be: Increased oxygen in the atmosphere is because of plants growing. Plants grow because rain provides them essential nutrients. A potential question would be: does rain cause plants to grow? The answer would be Yes. Another example narrative would be: Rain leads plants to grow. Plants growing causes less oxygen in the atmosphere. A potential question would be: does rain cause more oxygen in the atmosphere? The answer would be No. Another example narrative would be: The city's pollution levels increased because factories expanded their production. A separate heatwave occurred due to seasonal climate patterns, unrelated to factory activity. A potential question would be: did factory expansion cause the heatwave? The answer would be No."

### A.1.8   NARRATIVE + GRAPH PROMPT

"Use this narrative $N$ and this causal ordering $G'$ ((such that each item is a cause of every item after it, for example the first list item is a cause of the third, fourth, fifth items etc)) as context. Did $A$ cause $B$? Output your answer with $< answer > Yes/No < /answer >$. The cause can be direct or indirect."

### A.2   PARAMETRIC GRAPH EXPERIMENT

Let's call the graph of parametric knowledge $P$. We then take the odd indexed events (1st, 3rd etc) from $P$ and place them in the first half of the causal ground truth graph $G$ and the even indexed events (2nd, 4th etc) from $P$ in the second half of $G$. This process is shown in Figure 6.

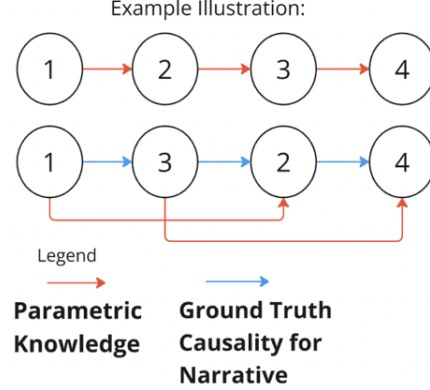

Figure 6: Example illustration (right) is of how $G$, the ground truth causality, is set up.

### A.3   COMPLEX GRAPH CREATION

To generate a ground-truth causal graph $G$ with rich structure (colliders, forks, and a spanning chain), for each choice of size $n$ we perform the following algorithm:

1. **Node sampling.** Draw $n$ distinct events

$$\{E_1, E_2, \ldots, E_n\} \subset \mathcal{E}$$

   uniformly at random without replacement.

2. **Determine motif counts.** (for $n \geq 4$)

$$k_{\max} = \lfloor n/2 \rfloor, \quad k_{\text{tot}} \sim \text{Uniform}(2, k_{\max}),$$

$$k_{\text{col}} \sim \text{Uniform}(1, k_{\text{tot}} - 1), \qquad k_{\text{fork}} = k_{\text{tot}} - k_{\text{col}}.$$

3. **Collider creation.** Repeat $k_{\text{col}}$ times:
   (a) Select two distinct "parent" nodes $p_1, p_2$ from those not yet used in any motif.
   (b) Select a "child" node $c$ that is neither $p_1$ nor $p_2$ and not yet used as a child.

(c) Add edges
$$p_1 \to c \quad \text{and} \quad p_2 \to c,$$
thereby forming a collider at $c$.

4. **Fork creation.** Repeat $k_{\text{fork}}$ times:
   (a) Select a "parent" node $p$ from those not yet used.
   (b) Select two distinct "child" nodes $c_1, c_2$ from the remaining unused nodes.
   (c) Add edges
   $$p \to c_1 \quad \text{and} \quad p \to c_2,$$
   forming a fork with shared parent $p$.

5. **Chain-connect remaining nodes.** Let $\mathcal{R}$ be the set of nodes not yet involved in any collider or fork.
   (a) Order $\mathcal{R} = \{r_1, \ldots, r_m\}$ arbitrarily, then add chain edges
   $$r_1 \to r_2, \ r_2 \to r_3, \ \ldots, \ r_{m-1} \to r_m.$$
   (b) To ensure the entire graph is connected, choose one node $u$ from among the previously used nodes (if any) and add
   $$u \to r_1.$$

## B   REAL-WORLD CAUSAL GRAPHS

### B.1   PROMPT TEMPLATES FOR NARRATIVE GENERATION

Recall that we have a ground truth causal chain graph of the form $V_1 \to V_2 \to \ldots \to V_N$ from *CauseNet* that we need to verbalize into a coherent narrative. For the semi-synthetic narratives, we use the LLM (GPT-4o) to do so one edge at a time, while ensuring that the newly verbalized edge logically follows the previous one. The following is the prompt template for generating the narratives in the topological order of the graph:

> Output a short narrative (use one or two sentences) that expresses the causal link $[V_i \to V_{i+1}]$ and logically follows this narrative:
> { Narrative for the previous edge $V_{i-1} \to V_i$}.
> Ensure that the combined sentences convey the causal chain $[V_{i-1} \to V_i \to V_{i+1}]$ and that the words $[V_i, V_{i+1}]$ are present. Only output the newly generated narrative.

Similarly, we generate narratives in the reverse topological order of the graph by verbalizing edges in the reverse direction with the following prompt template:

> Output a short narrative (use one or two sentences) that expresses the causal link $[V_i \to V_{i+1}]$ and logically follows this narrative:
> { Narrative for the previous edge $V_{i+1} \to V_{i+2}$}.
> Ensure that the combined sentences convey the causal chain $[V_i \to V_{i+1} \to V_{i+2}]$ and that the words $[V_i, V_{i+1}]$ are present. Only output the newly generated narrative.

For generating real-world narratives, for each edge $V_i \to V_j$, we use the set of sentences from *CauseNet*. Each edge in *CauseNet* is linked to multiple sentences from various sources. Picking a sentence for each edge at random and concatenating them does not always lead to sensible narratives. To improve the quality of narratives, we use the following prompt to concatenate sentences for adjacent edges:

> Consider the following sentences.
> { Sentence for edge $V_i \to V_{i+1}$ }. { Sentence for edge $V_{i+1} \to V_{i+2}$ }.
> Do the sentences logically follow each other and express the causal chain $[V_i \to V_{i+1} \to V_{i+2}]$? Answer with Yes or No.

For verbalizing narratives in the topological order, for a given graph $V_1 \to V_2 \to \ldots \to V_N$, we only use sentences such that the above prompt returns *Yes* for every pair of adjacent edges $V_i \to V_{i+1} \to V_{i+2}$. This ensures that the narrative as a whole remains coherent and conveys the entire causal chain graph. We use a similar prompting strategy to verbalize narratives in the reverse topological order.

## B.2 ELICITING PARAMETRIC KNOWLEDGE

We ask the LLM "Does $V_i$ typically have a causal (indirect or direct) effect on $V_j$?" and "Would it be atypical if $V_i$ had a (indirect or direct) causal effect on $V_j$?". If the LLM answers "No" and "Yes" to those respective questions, we would consider a causal relationship between $V_i$ and $V_j$ to contradict the LLM's prior knowledge that it learned from its pretraining corpora.

## B.3 SEMI-SYNTHETIC AND REAL-WORLD COMPLEX GRAPH ALGORITHM

Let $\mathcal{M} = \{(u,v)\}$ be the set of real-world causal edges from CauseNet. For each target size $n \in \{3, \ldots, 9\}$, we:

1. **Load CauseNet.**
$$\mathcal{M} \;=\; \big\{(u,v) \mid u \to v \text{ in CauseNet}\big\}.$$

2. **Extract collider and fork motifs.**
$$\mathsf{Colliders} = \{(p_1, p_2, c) \mid (p_1, c) \in \mathcal{M}, (p_2, c) \in \mathcal{M}, p_1 \neq p_2\},$$
$$\mathsf{Forks} = \{(r, c_1, c_2) \mid (r, c_1) \in \mathcal{M}, (r, c_2) \in \mathcal{M}, c_1 \neq c_2\}.$$

3. **Determine motif counts.**
$$\text{If } n = 3, \quad (k_{\mathrm{col}}, k_{\mathrm{fork}}) = \begin{cases} (1, 0) & \text{w.p. } 0.5, \\ (0, 1) & \text{w.p. } 0.5. \end{cases}$$

(for $n \geq 4$)
$$k_{\max} \;=\; \lfloor n/2 \rfloor, \quad k_{\mathrm{tot}} \sim \mathrm{Uniform}\big(2, k_{\max}\big),$$
$$k_{\mathrm{col}} \sim \mathrm{Uniform}\big(1, k_{\mathrm{tot}} - 1\big), \qquad k_{\mathrm{fork}} = k_{\mathrm{tot}} - k_{\mathrm{col}}.$$

4. **Select motifs.**
   - Sample $k_{\mathrm{col}}$ distinct triples from Colliders.
   - Sample $k_{\mathrm{fork}}$ distinct triples from Forks.

   Let $S$ be the union of all nodes appearing in these sampled triples.

5. **Pad or trim to size $n$.**
   - If $|S| > n$, uniformly subsample $n$ nodes from $S$.
   - If $|S| < n$, add random "seed" nodes (not already in $S$) until $|S| = n$.

6. **Build ground-truth edges $\mathcal{G} \subseteq S \times S$.**
   (a) *Colliders:* for each $(p_1, p_2, c)$ chosen, add $p_1 \to c$ and $p_2 \to c$.
   (b) *Forks:* for each $(r, c_1, c_2)$, add $r \to c_1$ and $r \to c_2$.
   (c) *Chains:* for any remaining $(u, v) \in S \times S$ with $(u, v) \in \mathcal{M}$ and neither $u$ nor $v$ used in the above, add $u \to v$ to ensure connectivity.

7. **Narrative generation.** For each $(u \to v) \in \mathcal{G}$:
   For the semi-synthetic case - prompt the LLM to generate a sentence linking u to v using the forward topological ordering prompt.
   For the real-world case: Find a causal sentence linking u and v in the Cause-Net database

## C REAL-WORLD COMPLEX GRAPH CREATION

### C.1 ADDITIONAL EXAMPLES OF THE GENERATED NARRATIVES

#### C.1.1 SEMI-SYNTHETIC NARRATIVES

Below, we present some examples of semi-synthetic narratives in the forward and reverse directions.

The narrative in the forward direction for the chain *higher prices → reduced demand → lower prices*:

> As *higher prices* swept through the market, consumers began to tighten their budgets, leading to a noticeable *reduction in demand* for many goods. As a result of the *reduced demand*, suppliers were forced to *lower prices* in order to attract buyers back to the market.

The narrative in the reverse order for the causal chain *bankruptcy → bad credit → rejection → anger*:

> The sting of rejection ignited a fire within her, transforming her hurt into a seething anger that demanded to be felt. Her bad credit had led to the rejection she never saw coming, and now that sting of rejection ignited a fire within her, transforming her hurt into a seething anger that demanded to be felt. Her bankruptcy had left her with bad credit, a shadow that loomed over her every application, and now that sting of rejection ignited a fire within her, transforming her hurt into a seething anger that demanded to be felt.

The narrative in the reverse order for the causal chain *pollution → climate change → extreme weather events → natural disasters*:

> As extreme weather events become more frequent and severe, they increasingly lead to devastating natural disasters that disrupt communities and ecosystems alike. Climate change is driving the rise in extreme weather events, which in turn are causing unprecedented natural disasters that threaten the stability of communities and the health of ecosystems. Pollution is a major contributor to climate change, which is driving the rise in extreme weather events that threaten the stability of communities and the health of ecosystems.

#### C.1.2 REAL-WORLD NARRATIVES

Below, we present some examples of real-world narratives in the forward and reverse directions.

The narrative in the forward direction for the chain *higher prices → reduced demand → lower prices*:

> *Higher prices* generally lead to reduced demand. *Lower prices*, caused by *reduced demand* and increased competition for soybeans and corn, largely contributed to the overall bulk export decline.

The narrative in the reverse order for the causal chain *bankruptcy → bad credit → rejection → anger*:

> Embittered by an abusive upbringing, seething with resentment, irritated by others' failure to fulfill his or her superior sense of entitlement, and fuelled by anger resulting from rejection, the serial bully displays an obsessive, compulsive and self-gratifying urge to displace their uncontrolled aggression onto others whilst exhibiting an apparent lack of insight into their behavior and its effect on people around them. Bad credit normally leads to rejection but now with bad credit secured loan, you can avail the loan of your choice. For example, if you are applying for a loan, the lender may reject your application on the basis of bad credit caused by bankruptcy.

The narrative in the reverse order for the causal chain *pollution → climate change → extreme weather events → natural disasters*:

In addition to forced migrations from rising seas, climate change is also increasing extreme weather events causing natural disasters such as cyclonic storms (hurricanes or typhoons), floods and droughts. This is worsened by extreme weather events caused by climate change. This landmark bill would jump start the economy by creating millions of new clean energy jobs, increase national security by reducing dependence on foreign oil, and preserve the planet by reducing the pollution that causes climate change.

## C.2 PROMPT TEMPLATES FOR ASSESSING CAUSAL REASONING

We use the following template for the Direct prompting strategy:

Consider the following hypothetical narrative.

{narrative}

According to the hypothetical narrative, does {cause} have a (direct or indirect) causal effect on {effect}? Answer in Yes/No.

We use the following template for the Chain-of-Though (CoT) prompting strategy:

Consider the following hypothetical narrative.

{narrative}

According to the hypothetical narrative, does {cause} have a (direct or indirect) causal effect on {effect}? Think step-by-step and end your answer with <answer>Yes/No</answer>.

We use the following template to extract a chain graph from the narrative:

Consider the following hypothetical narrative.

{narrative}

According to the hypothetical narrative, construct a causal chain graph using the following nodes: { nodes in random order }. Ensure that the graph contains all the given nodes and only output a single chain graph of the form <graph>node1 → node2 → node3 </graph>. Only output the graph between the <graph></graph>tags.

## C.3 NECESSARY COMPUTE

No pretraining was done so no GPUs were needed. We used cloud based API calls to pre-trained models like ChatGPT, Anthropic and Llama. We estimate that for the synthetic portion, our API calls to ChatGPT, Anthropic and LLama took 10 hours each. For the semi-synthetic and real-world portion, we had roughly 10 hours of API calls for ChatGPT and Llama each. So in total, roughly 50 hours of API usage. As the majority of the computational burden fell on cloud based API calls, no significant CPU resources are required either.

# D ADDITIONAL RESULTS - SYNTHETIC DATA

## D.1 FORWARD VS REVERSE EXPERIMENTS

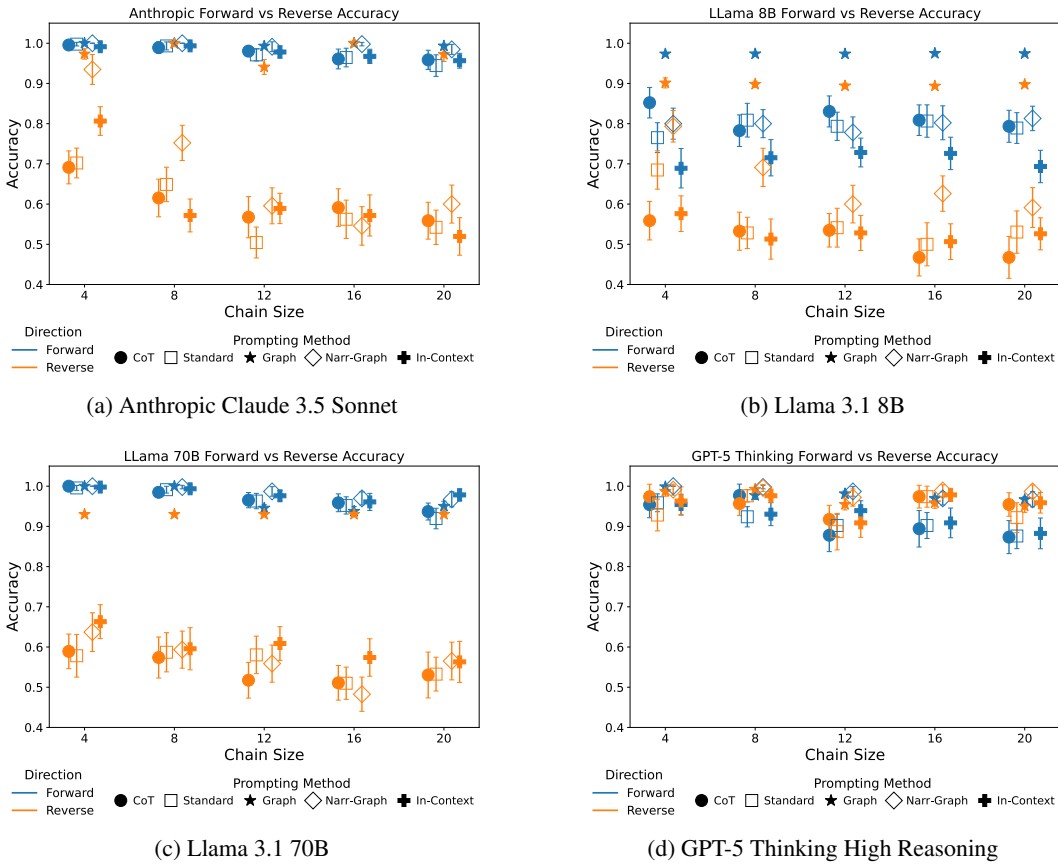

(a) Anthropic Claude 3.5 Sonnet

(b) Llama 3.1 8B

(c) Llama 3.1 70B

(d) GPT-5 Thinking High Reasoning

Figure 7: (a) Anthropic Claude 3.5 Sonnet, (b) LLama 3.1 8B, (c) LLama 3.1 70B and (d) GPT-5 Thinking High Reasoning Test of the LLM's ability to reason on narratives written in the Forward and Reverse topological orientations. Chain size is the number of nodes in ground truth $G$. The "Graph" prompting method uses only the extracted graph $G'$ to reason, "Narr-Graph" uses both the narrative and extracted graph, and "Standard, CoT, In-Context" all use only the narrative. Accuracy measures LLM answer agreement with $G$. The points in the graph are represented with a slight horizontal stagger around the relevant chain sizes (4,8,12 etc) for ease of visual understanding. We show a 95% CI.

In these graphs, we perform the Forward vs Reverse Experiments for (a) Anthropic Claude 3.5 Sonnet, (b) LLama 3.1 8B, (c) LLama 3.1 70B and (d)GPT-5 Thinking High Reasoning. Across a scale of model sizes and reasoning capabilities, patterns emerge. We see that a consistent failure mode remains of models (small or large) being much worse at reasoning about reverse narratives than ones in the forward direction – until we get to the reasoning model which closes the gap. We also notice that the reasoning model doesn't score perfectly in the forward regime like many of the non-reasoning models. The fact that it makes some mistakes in that regime, while still doing well, is indicative of actual reasoning and not following a simple shortcut.

## D.2 CAUSAL VS ANTI-CAUSAL EXPERIMENTS

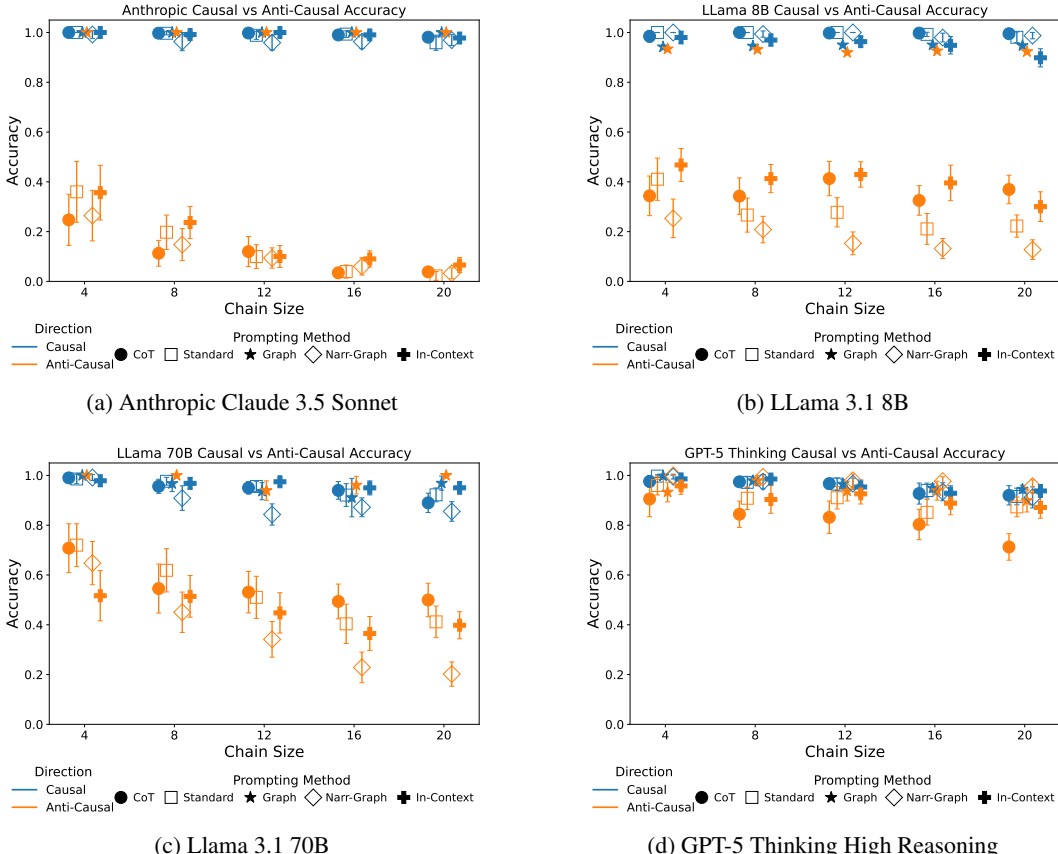

Figure 8: (a) Anthropic Claude 3.5 Sonnet, (b) LLama 3.1 8B, (c) LLama 3.1 70B and (d) GPT-5 Thinking High Reasoning Test of the LLM's ability to reason on narratives that agree with parametric knowledge (Causal) and disagree with parametric knowledge (Anti-Causal). 95 % CI is shown.

In these graphs, we perform the Causal vs Anti-Causal Experiments for (a) Anthropic Claude 3.5 Sonnet, (b) LLama 3.1 8B, (c) LLama 3.1 70B and (d)GPT-5 Thinking High Reasoning. We see that larger models like GPT-4o and Claude 3.5 Sonnet perform far worse on knowledge that conflicts with their pre-training compared to LLama models, possibly because they have been trained on so much more data than the LLama models. As such, we can say that size of the model doesn't necessarily translate into better performance for the failure modes we identified. What does seem to translate into significantly better performance is the amount of reasoning capability the model explicitly has.

## D.3 COMPLEX VS SIMPLE GRAPHS

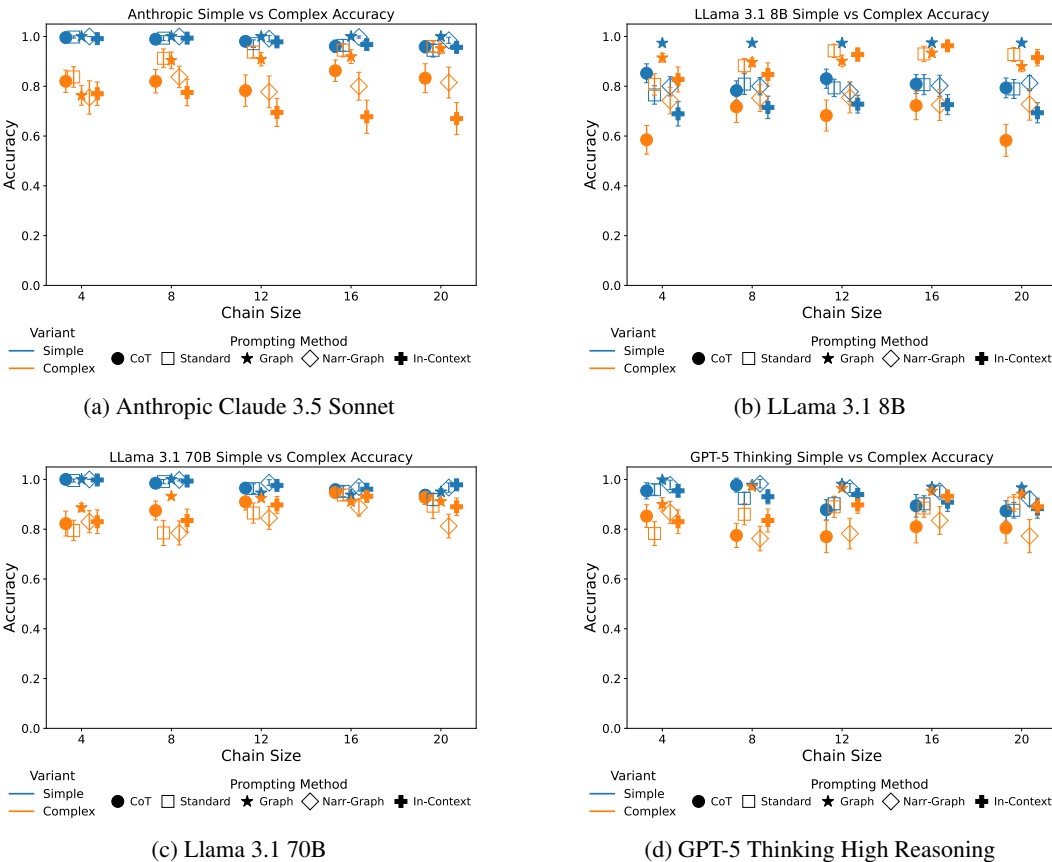

Figure 9: (a) Anthropic Claude 3.5 Sonnet, (b) LLama 3.1 8B, (c) LLama 3.1 70B and (d)GPT-5 Thinking High Reasoning Test of the LLM's ability to reason on narratives generated from Complex graphs as opposed to Simple chain graphs. 95 % CI is shown.

In these graphs, we perform the Complex vs Simple Experiments for (a) Anthropic Claude 3.5 Sonnet, (b) LLama 3.1 8B, (c) LLama 3.1 70B and (d)GPT-5 Thinking High Reasoning. We see relatively similar performance across all models except for Llama 3.1 8B which has more variable performance. It's general inconsistency may be due to the fact that it is a weaker model than the others presented.

## D.4 GRAPH EDIT DISTANCE

Here in Table 2, we compute the average graph edit distance per, using the implementation from the networkX package in python (Abu-Aisheh et al. (2015)) where they define graph edit distance as "It is defined as minimum cost of edit path (sequence of node and edge edit operations) transforming graph G1 to graph isomorphic to G2." 95% CIs given. We find GED to be calibrated for a different measure than graph accuracy, as for example one small change in the causal graph with GED drastically impact accuracy – so two structures with similar GEDs can have drastically different accuracies.

| Models | Graph type | | | | |
|---|---|---|---|---|---|
| | Forward | Reverse | Causal | Anti-Causal | Complex |
| LLaMA 3.1 8B | .04(.04,.04) | .08(.08,.08) | .03(.03,.03) | .03(.03,.03) | .08(.07,.11) |
| LLaMA 3.1 70B | .04(.04,.04) | .07(.07,.07) | .02(.02,.02) | .02(.01,.02) | .07(.06,.09) |
| Claude 3.5 Sonnet | 0(0,0) | .02(.01,.02) | 0(0,0) | 0(0,0) | .08(.07,.11) |
| ChatGPT-4o | .03(.03,.03) | .03(.02,.05) | .13(.12,.15) | .09(.08,.11) | 0(0,0) |
| ChatGPT-5 (Thinking) | .02(.02,.02) | .03(.03,.03) | .02(.02,.02) | .04(.03,.05) | .04(.03.04) |

Table 2: Graph Edit Distance by graph type for different models.

# E    ADDITIONAL RESULTS - SEMI-SYNTHETIC AND REAL WORLD DATA

## E.1    FORWARD VS REVERSE LLAMA

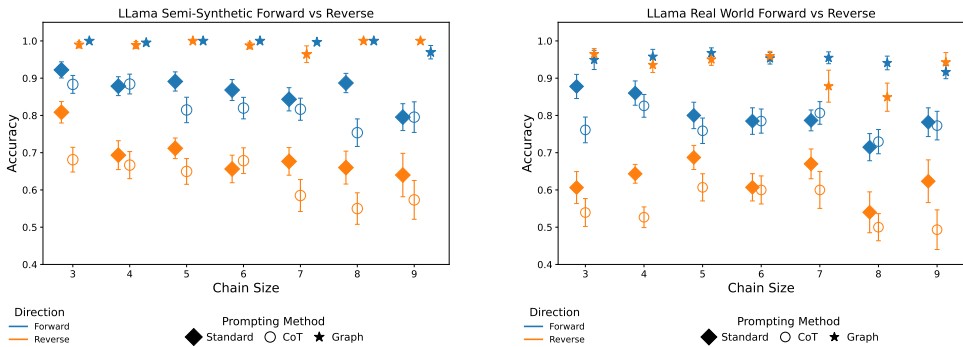

Figure 10: (LLama 3.1 8B) The accuracy of various prompting strategies (error bars denote 95% CIs) in the Semi-Synthetic and Real-World Regimes using CauseNet.

We observe that the accuracy is lower in the reverse direction in both regimes, and slightly lower yet in the real world regime. This is consistent with previous findings. The extracted graph does well.

## E.2    PARAMETRIC EXPERIMENT LLAMA

|  | Standard | CoT | Graph |
|---|---|---|---|
| **Semi-synthetic** | | | |
| Without Conflict | 88.4 | 83.7 | 99.5 |
| With Conflict | 61.4 | 57.9 | 98.2 |
| **Real-world** | | | |
| Without Conflict | 81.6 | 79.2 | 95.1 |
| With Conflict | 48.8 | 49.9 | 93.2 |

Table 3: (LLama 3.1 8B) The average accuracy across different narratives with the three prompting strategies partitioned by whether the cause-effect pairs conflict with the LLM's parametric knowledge (we omit the 95% CIs as they are smaller than 0.3).

We observe that the accuracy is drastically lower with conflicting information in both regimes, and slightly lower yet in the real world regime. This is consistent with previous findings. We again see the graph doing very well.

## E.3 SIMPLE VS COMPLEX LLAMA

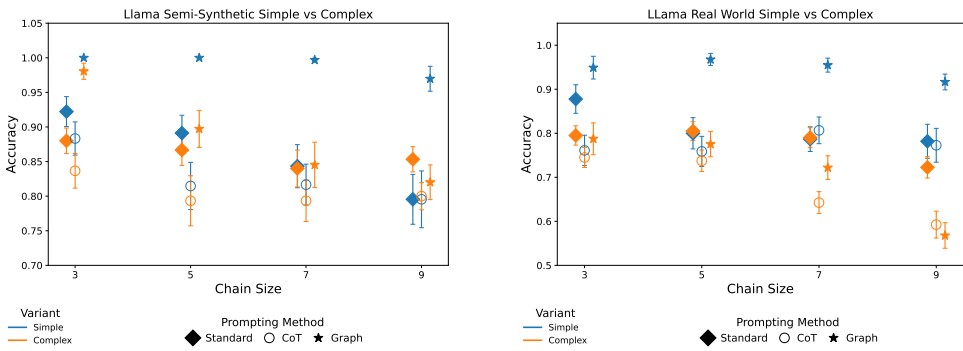

Figure 11: (LLama 3.1 8B) accuracy on narratives generated from Complex graphs as opposed to Simple chain graphs for semi-synthetic narratives (left) and real-world narratives (right). 95 % CI is shown.

We see slight degradation of accuracy in the complex regime as opposed to the simple one, with the graph not fully recovering accuracy in the complex regime. This is consistent with previous findings.

## F    Human evaluation protocol and results

We conducted a human evaluation of narrative quality and graph faithfulness across the three regimes (synthetic, semi-synthetic, and real-world). For each regime, we sampled 150 unique narratives, yielding a total of 450 narratives. Three external graduate-student reviewers (none of whom are authors) served as annotators.

For each regime, the 150 narratives were partitioned into three disjoint batches of 50 narratives. Each batch was assigned to a distinct pair of reviewers, so that every narrative was independently annotated by exactly two reviewers and each reviewer annotated 100 narratives per regime (300 narratives in total across all three regimes).

**Tasks.**    For each narrative, annotators were given the list of node names and asked to:

1. Reconstruct the causal chain graph $G'$ that they believed the narrative implied, and

2. Provide a binary judgment of whether the narrative was *fluent and coherent*.

We say that a narrative is *exactly reconstructed* if the reconstructed graph $G'$ matches the ground-truth graph $G$ exactly (all nodes present, all directions correct, and no extra edges). We report two notions of reconstruction accuracy:

- **Narrative-level accuracy**: fraction of narratives in a regime for which *both* annotators exactly reconstructed $G$.

- **Label-level accuracy**: fraction of individual annotations (over all annotator–narrative pairs) that exactly reconstructed $G$.

For agreement, we compute (i) the percentage of narratives for which the two annotators produced identical graphs, and (ii) Cohen's $\kappa$ on the space of complete graph structures (treating each distinct chain as a categorical label).

**Reconstruction accuracy and agreement.**    Table 4 summarizes reconstruction performance and annotator agreement. Across all regimes, narrative-level reconstruction accuracy is at least 97%, and label-level accuracy is close to $98\%\,\check{}\,99\%$.

| Regime | # narr. | # ann. | Exact (N, %) | Exact (L, %) | Agree (%) | $\kappa$ |
|--------|---------|--------|--------------|--------------|-----------|----------|
| Synthetic | 150 | 300 | 98.0 | 98.7 | 98.7 | 0.95 |
| Semi-synthetic | 150 | 300 | 97.3 | 98.3 | 98.0 | 0.93 |
| Real-world | 150 | 300 | 97.3 | 98.0 | 98.7 | 0.91 |

Table 4: Human evaluation of graph reconstruction. "Exact (N, %)" is the fraction of narratives for which both annotators exactly reconstructed the ground-truth graph $G$. "Exact (L, %)" is the fraction of individual annotations that exactly reconstructed $G$. "Agree (%)" is the fraction of narratives for which the two annotators produced identical graphs (whether correct or incorrect). $\kappa$ denotes Cohen's kappa, a chance-corrected measure of inter-annotator agreement computed over the space of complete graph structures.

**Fluency judgments.**    For fluency, we summarize ratings at the narrative level. A narrative is counted as *fluent (both)* if both annotators marked it fluent; it is counted as *fluent ($\geq$1)* if at least one annotator marked it fluent. Cohen's $\kappa$ is computed on the binary fluent/non-fluent labels.

As shown in Table 5, the vast majority of narratives in all regimes are judged fluent, with slightly lower fluency rates in the real-world regime where source sentences can be more heterogeneous.

| Regime | Fluent (both, %) | Fluent ($\geq$1, %) | Cohen's $\kappa$ (fluency) |
|---|---|---|---|
| Synthetic | 96.0 | 99.3 | 0.88 |
| Semi-synthetic | 94.0 | 98.7 | 0.86 |
| Real-world | 92.0 | 97.3 | 0.84 |

Table 5: Fluency judgments across regimes. "Fluent (both)" counts narratives where both annotators judged the narrative fluent; "Fluent ($\geq$1)" counts narratives where at least one annotator judged the narrative fluent.

Overall, human auditors almost always reconstruct the correct causal chain from our narratives and judge them to be fluent, suggesting that our narrative generation procedures produce text that is both faithful to the underlying graph $G$ and natural to read.

