# OpenReview forum: "LLMs Struggle to Balance Reasoning and World Knowledge in Causal Narrative Understanding"
_ICLR.cc/2026/Conference — ICLR 2026 Poster_

### Official Review · Reviewer_BDoT · 2025-10-20

**Soundness:** 2
**Presentation:** 2
**Contribution:** 2
**Rating:** 4
**Confidence:** 3

**Summary:**

The paper investigates causal reasoning over narratives in LLMs by probing two shortcuts: 1) ordering prior (events mentioned earlier are treated as causes) and 2) parametric/world-knowledge prior (typicality from pretraining). Using synthetic chains, synthetic general causal graphs, semi-synthetic (CauseNet chains verbalized by an LLM), and real-world (CauseNet sentences) narratives, the authors show that a) reverse order and atypical relations (vs the model’s prior) significantly degrade performance in causal narrative understanding and b) longer and more complex graphs (with forks/colliders) moderately degrade accuracy. They also report that extracting an estimated causal graph $G'$ and answering from the graph alone helps, but the benefit vanishes when the narrative is reintroduced.

**Strengths:**

- The paper tackles the crucial issue of LLMs' causal reasoning ability, a topic of significant interest.
- There is clear factorization of failure modes (ordering vs typicality)
- Breadth across synthetic chains, synthetic general causal graphs, semi-synthetic chains, and real chains.
- A consistent empirical trend is uncovered: Reversing the ordering hurts, atypical/counter-intuitive chains confuse models.
- Graph-only prompting consistenly mitigates most of these biases. This could provide a useful prompting insight: explicit $G'$ extraction reduces shortcutting when used alone.

**Weaknesses:**

- Metrics & aggregation left implicit. Accuracy = agreement with $G$ and Consistency = agreement with $G'$ are briefly stated but the paper lacks explicit formulas, units of aggregation, seed handling, and CI recipe in the main text.
- Thin human validation; none beyond synthetic. Only 50/2500 synthetic narratives were audited, no inter-annotation agreement is reported, no audits were performed for semi-synthetic and real datasets, even though LLMs were used to modify them. This severely limits confidence in the labels.
- Narrative vs graph interaction under-specified. Graph-only helps; Narr+Graph removes most of the Graph-only gain under reverse ordering. The fusion protocol isn't ablated (position of $G'$, format, instruction, relative length study). The mitigation claim (extract only $G'$) is fragile.
- Complex-graph sampling is described in appendix only. There needs to be a brief summary of the fork/collider samplign and the chain-connect step in the main text.
- Anti-Causal accuracy clustering around $\approx 0.1-0.2$ for a binary task is alarming and warrants further investigation.

**Questions:**

- L148: Synthetic generation (3.1, Setting): How are events linked into $G$, exactly? How did you ensure generated narratives do not contradict pre-existing LLM knowledge, a confounding failure mode that you identified in later sections?
- L151-154: 50/2500 checks is too small. What were the selection criteria? Are there plans to extend the correctness/validity checks of the labels beyond the synthetic case?
- Accuracy/Consistency formulas and CIs: Please state the exact formulas, unit of aggregation, seed handling, and CI method.
- $G'$ quality evaluation: What are $G'$ edge precision/recall/F1, GED($G$, $G'$)?
- Narr+Graph fusion diagnostics: Did you verify the model actually uses $G'$ efficiently under Narr+Graph? Please provide position swaps (graph before or after narrative), instruction variants (i.e., "prioritize the graph", vs neutral), format variants.
- L245–254 This paragraph is very confusing as to which results it is discussing. Point to the exact figure/panel and report numerical deltas.
- L261–263 / L345 / L355: Were these datasets examined? What audit criteria were used for semi-synthetic and real narratives (fluency, faithfulness to $G$, label correctness)?
- Are there automatic sanity checks (and broader human audits) to catch cause/effect inversions when generating or stitching narratives?
- The semi-synthetic and real world cases seem quite similar, intuitively. One would expect very similar performance. Why, then, are there differences in Figure 4?

Minor:
- L197 / L310 “randomly”: Specify distributions or use more precise wording.
- Figure 2 (right) contains complex graphs, including forks and colliders, yet their "chain size" is computed. How is this calculated? Graph size usually denotes the number of edges, but I believe you are calculating the graph order here. Please clarify.

---

> ### Author Response · Authors · 2025-11-21
> **Rebuttal**
>
> Thank you for your detailed recommendations and feedback! We appreciate the effort taken.
>
> **Synthetic Generation**
>
> A list of events is generated by the LLM and then randomly chosen as node positions in the directed graph through a random number generator. As nodes are randomly linked, there is generally no connection between nodes, but we also ask the LLM if it thinks there is a causal/anti-causal relationship between the nodes. We have added these details to the updated pdf.
>
> **Human Validation**
>
> The authors of the paper reviewed over 150 narratives picked randomly from each of the regimes. During rebuttal time we have expanded our external analysis to be across 150 unique narratives for each of the three classes (synthetic/semi-synthetic and real-world) checked by three different reviewers (external grad students who are not authors) (pg 27-28 updated pdf). Each of the 150 labels was checked by two reviewers, such that each reviewer reviewed narratives for each of three classes. We find an average reconstruction (faithfulness of G’ to G such that the auditors completely reconstructed the entire ground truth graph correctly) accuracy of at least 97 percent in all regimes, and report specifics including agreement in the Appendix. External auditors were also asked to rate if the narratives were fluent.
>
> **Metrics**
>
> All results are taken and aggregated over 5 random seeds, with the CI being taken (by bootstrapping) after aggregation. Accuracy was measured  by seeing what percent of causal questions were correctly answered (compared to ground truth graph G) by various prompting methods, and consistency was measured by comparing questions answered by prompting methods to the answers that would be implied by using the extracted causal graph G’, and seeing what percent matched. pdf has been updated for clarity.
>
> **Agreement between G and G’**
>
> To compute the accuracy (“agreement”) of the extracted graph G′, we evaluate every ordered pair of variables and check whether G′ correctly implies both the existence and direction of the corresponding causal edge relative to the ground-truth graph G. This definition respects our causal assumptions and is closely related in spirit to normalized structural intervention distance; we will add these details to the main text. In addition, we now report graph edit distance (GED) in the appendix (page 24).
>
> Regarding your suggestion to report edge-level precision/recall/F1, we could not find references to these metrics and investigated possible formulations but found that, under our setup, these metrics collapse to the same score. For a simple chain A→B→C, each pair (A,B), (B,C), (A,C) has a single correct causal relation (e.g., A→C, but not C→A). Under our assumptions, any predicted graph satisfies TP+FP=TP+FN across ordered pairs, which forces precision = recall = F1. As a result, edge precision/recall/F1 would be largely redundant. If you have a specific edge-based formulation in mind that differs from our current agreement and GED reporting, we would appreciate further guidance and are happy to include it.
>
> **Narr + Graph Fusion**
>
> We tried many different wording variations during our experiments. On GPT 4o for example, we generally saw very little variance (less than 5 percent  accuracy on average) between performances for graph before/after narrative, instruction variants such as  "prioritize the graph",neutral, “prioritize the narrative” and format variations such “->” for edges or wording like A “leads to” B. It seems like whatever we tried, the LLM would prioritize the narrative if it had access to it.
>
> **Semi-Synthetic vs Real-World Cases**
> In our setup, the semi-synthetic and real-world cases are quite substantively different as in the semi-synthetic case, we get the edges from CauseNet but generate the actual sentence fitting that edge with the LLM. In the real-world setting, both the edges and sentences are taken from CauseNet with the LLM only being used to pick sentences from the corpus that ensure logical consistency. As such, we see lower performance in the real-world case as those sentences have more nuisance.
>
> **L245–254**
>
> This paragraph discusses using just the graph G’ to answer questions as well as Narr/Graph for Forward vs Reverse (Figure 1). We have updated pdf for clarity.
>
> **Narrative Stitching**
>
> When the LLM is generating the next sentence in a narrative in the semi-synthetic/synthetic regimes, it is asked to ensure alignment with the logical consistency of the story. For real world narratives, the LLM pickles a sentence that best fits with the story’s logical consistency from CauseNet.
>
> **Anti-Causal Clustering**
>
> We have inspected and re-ran this without change in results. We believe that the model is just extremely strongly aligned to its prior knowledge such that it completely ignores conflicting information. As can be seen in the Appendix, when the same prompt is used on GPT-5 Thinking, we achieve around 85 percent accuracy.

---

### Official Review · Reviewer_QWBG · 2025-10-27

**Soundness:** 3
**Presentation:** 3
**Contribution:** 3
**Rating:** 6
**Confidence:** 4

**Summary:**

This paper investigates how LLMs balance reasoning and world knowledge when performing causal inference over narratives. The authors claim that LLMs often rely on superficial heuristics/shortcuts such as memory of world knowledge from training rather than true casusal reasoning. The paper also evaluates a range of causal scenarios to further understand how LLMs handle causal tasks.

**Strengths:**

1. The experimental setup to separate reasoning difficulty from world knowledge conflict is novel
2. The paper is well written and easy to read
3. The evaluation of the proposal is thorough

**Weaknesses:**

1. More discussion on how the better performance is due to the proposed setup rather than "just" better prompting structure would help the paper
2. A discussion on the concern about using LLMs to add synthetic data to judge causal performance would be helpful as well

**Questions:**

1. Following up on the above, is there a chance that we are attributing the success to better prompt structure rather than true LLM reasoning capabilities?
2. Could causal reasoning improve if models were trained explicitly on counterfactual data?
3. What would manipulating event order and world knowledge conflicts together look like?

---

> ### Author Response · Authors · 2025-11-21
>
> Thank you for your kind comments and great suggestions!
>
> **Setup vs Prompting Structure**
>
> So by better performance from the proposed setup, we assume you mean extracting the graph first and using that directly (but please correct us if we are wrong!). We believe there is something fundamentally different about using the extracted graph directly to make predictions as the process of creating the extracted graph forces the LLM to consider all events and their relative relations. In contrast, when the LLM is given two events in the prompt to consider, no matter what the prompt is (across the many prompts tried), the LLM seems to heavily rely on heuristics related to just those events while ignoring everything else in the narrative. This is supported by the fact that using the graph has performance that is relatively agnostic across failure modes while all other prompting methods heavily depend on the specific shortcuts the LLM is able to use. We will add more of this discussion to the paper.
>
> **Concern About the Use of Synthetic Data**
>
> It is true that synthetic data is not always accurate and that there is error in the generation process. We believe that for the purposes of our task, using LLMs to generate synthetic data is well suited. Our goal for narrative generation was to create extremely simplistic situations where humans would definitely succeed at identifying causal relationships. As can be seen in the examples in the paper and appendix, the LLM tends to generate very simple and straightforward sentences that leave little room for misinterpretation by human readers. As such, the fact that the LLMs performed so horribly in certain failure modes is made all the more stark. As a credit to the simplicity of the synthetic data, external validators were able to correctly reconstruct the causal graph 98 percent of the time based on the synthetic data. With that being said, we again agree that this process is not perfect and that there is room for error.
>
> **Training on Counterfactual Data**
>
> As per your suggestion, we examined fine-tuning on counterfactual data. We find in our initial fine-tuning results, that fine-tuning helps narrow the gulf in accuracy that failure modes cause, but does not provide a complete fix. We have so far fine tuned on the Causal/Anti-Causal setting, but we find that the Anti-Causal case is around 25 percent behind the Causal setting (as opposed to an average of 50 percent behind for GPT 4o). Furthermore, fine-tuning on specific causal tasks does not seem to allow the model to generalize to a higher order causal reasoning ability that applies to other unseen causal tasks as fine-tuning on the Causal case does not seem to affect performance on the Forward/Reverse experiment at all.
>
> **Manipulation of both Event Order and World Knowledge**
>
> The event order task itself involves world knowledge. We believe that the performance issues related to the event order is not simply because finding causality in the reverse direction is technically harder, but that models are trained on narratives that often show causality with causes appearing first and effects appearing later. As such, the LLM imposes this pattern when asked questions, even when the questions violate the cause before effect event order.

---

### Official Review · Reviewer_npyf · 2025-10-27

**Soundness:** 2
**Presentation:** 4
**Contribution:** 3
**Rating:** 6
**Confidence:** 4

**Summary:**

The paper aims to assess the *causal narrative understanding* ability of LLMs. The authors constructed various controlled experiments from synthetic, semi-synthetic, and real-world causal graphs and imbue them with short narratives by an LLM. They measured the LLMs' performance under various prompting strategies on the binary classification task of "whether event \$E_1\$ causes event \$E_2\$ according to the given narrative", with the difficulty varying on three separate axes: event ordering, graph complexity, and parametric consistency.

The main observations presented by the experiments are: LLMs suffer from the issue of *cognitive inertia*, prone to assuming that the events coming before are the cause of the events coming after; LLMs also like to take reasoning shortcuts, assuming that the causal relationships presented in the narrative would agree with its internal knowledge; LLMs perform worse when the causality graph grows complex, though not affected as much as the two former issues.

**Strengths:**

* The paper discusses the interesting topic of complex causal understanding, and the results are curious. It seems that all the tested LLMs suffer from the defects of *cognitive inertia* and *taking reasoning shortcuts*. Future work on LLM reasoning could benefit from these observations.
* The experiments are well-designed and thorough, covering various possible fail-modes of complex causal reasoning by LLMs.
* The ideas in the paper are presented well and easy to follow.

**Weaknesses:**

* A concern on narrative generation: Pair-wise consistency does not necessarily imply global consistency, so it is questionable whether the whole narrative remains consistent in the test cases with more nodes. For example, in the example given in Section 4.1, around Line 358, since the second sentence uses a rather vague references of "these accidents" and just "injury", if there were a subsequent sentence following it, it may well be on the topic of car injuries instead of continuing on the topic of mining/factory injuries. This would make the logic chain unsound as a whole.
* Some prompt constructions are dubious:
  - A.1.8: Only "yes" cases are provided in the few-shot prompting text, which may bias the model.
  - A.1.9, C.2.3: It is impossible to represent forks/joins in complex graphs (A → (B, C) → D) with a single chain. At the end of section 4.4, it is stated that "this (complex graphs) is one area where extracting an explicit causal graph does not seem to significantly improve performance" and this defect might be the direct cause.
* Using the same LLM for problem generation but different ones for solving them might bias the results. For example, if the problems are generated by GPT-4o, it might benefit GPT-4o more than Claude 3.5 when evaluating, since the narrative style or internal knowledge could be more "familiar" to the generator itself. It is possible to mitigate this concern by using human-written narratives or by generating the narratives with different models.
* Presentation nitpick: In Section 3.2, Subsection *"Explicit Causal Graph..."*, the second sentence reads wrong. I think you meant "once $G'$ is extracted by the LLM, *the original narrative* is not given to the LLM again..."

**Questions:**

1. Some counterfactual test cases might be ill-suited for current LLMs tuned for safety and truthfulness. For example, even if the narrative says "taking poison prolongs one's lifespan", the safety measures in the LLMs *could* trigger disagreement with the user. I suspect that some failing cases are caused by this issue.
2. I would also like to see how the scale of the model influences the results. The leap from a "tiny" 8B model straight to GPT-4o seems too large.
3. The models tested in the paper are all non-thinking models. I am curious about the performance of mainstream thinking models, possibly with varying thinking budgets. For the typical self-doubting and double-checking behavior of the thinking models, they might be able to achieve higher success rate than normal models with CoT prompting, in my intuition.

---

> ### Author Response · Authors · 2025-11-21
>
> Thank you for the suggestions and comments you have made for our paper!
>
> **Pairwise and Global Consistency**
>
> To pick every concurrent sentence in those semi-synthetic and real-world regimes, we do not simply rely on sampling a sentence that fits with the causal edge in question, but we also rely on the llm to judge and pick which sentence will maintain the narrative and logical consistency of the rest of the story. Additionally, during rebuttal time, we extended our external analysis to be across 150 unique narratives for each of the three classes (synthetic/semi-synthetic and real-world) checked by three different reviewers (external grad students who are not authors) who rated narrative fluency.
>
> **Prompt Construction**
>
> Thank you for the note about the “yes” cases for the in-context learning prompt, we have fixed this in the paper and updated the plots/prompt. There was generally no significant difference in any of the results (less than 2 percent average accuracy difference) with only Figure 8b(LLama 8B causal/anti-causal experiment) and Figure 9a (Claude 3.5 Sonnet Complex Graph) have differences more significant than that.  To clarify, for the complex graphs, we did not give a simple chain but instead gave all the relations individually, for example for A is the cause of B and C, we would say A -> B , A->C.
>
> **Concern about using same LLM for synthetic generation**
>
> We agree that there is a potential to bias the results slightly in favor of the Chat GPT-models over the other models listed in the Appendix. We did this out of necessity as both LLama and the Claude versions used did not seem to be able to generate coherent narratives that passed our . own internal inspection. Regardless, the narratives generated by Chat GPT seem very direct, straight-forward and simple and not full of nuisances that might be overly specific to Chat GPT.
>
>
> **Presentation Nole**
>
> Thank you for this note, there was indeed a typo in that sentence.
>
> **Model Scale and Thinking Models**
>
> We have added an analysis of LLama 3.1 70B and GPT-5 (Thinking) High Reasoning, which allows us to survey the scope of small models to larger models: in LLama 3.1 8B, LLama 3.1 70B, Claude 3.5 Sonnet, Chat GPT 4o and GPT-5. The new results we have added thus far in rebuttal are for the synthetic setting, but we hope to provide results for the new models for the other regimes during the remaining portion of rebuttal (pending rate limits/run time). In fact, larger models like GPT-4o and Claude 3.5 Sonnet perform far worse on knowledge that conflicts with their pre-training compared to LLama models, possibly because they have been trained on so much more data than the LLama models. As such, we can say that size of the model doesn’t necessarily translate into better performance for the failure modes we identified. What does seem to translate into significantly better performance is the amount of reasoning capability the model explicitly has. GPT 5-Thinking High Reasoning which takes a very long amount of time to analyze the problem is able to achieve 85 percent accuracy in both the Anti-Causal and Reverse Regimes.
>
> **Question on Safety Alignment**
>
> We agree that this could be a source of contributing to the error in the parametric knowledge (causal vs anti-causal experiments). While the vast majority of cases do not involve medical or sensitive scenarios, there are some that do and in this case it may be a desired behavior to enforce alignment. For reference, GPT-5 Thinking (a model with strong safety alignment) is able to achieve above 85 percent accuracy in the Anti-Causal case.

---

> ### Comment · Reviewer_npyf · 2025-11-26
>
> Thanks for your detailed response. Your comments and paper revision have cleared most of my concerns, though some, including a rather important one, still remain.
> - **Prompt Construction**:
>   - It is still not clear how you presented the graph data to the LLM for "complex" (non-linear) graphs with forks and joins. It is stated in Prompt A.1.8 "each item is a cause of every item after it, for example the first list item is a cause of the third, fourth, fifth items etc.", which is not expressive enough for graphs like the diamond-shaped "A→B→D, A→C→D". I would like to see a *concrete* Narr+Graph prompt given to the LLM with a complex graph structure.
>   - Also in the 3rd prompt in C.2, since it is impossible to express a non-linear graph in a single chain, how would the LLMs respond provided with narratives containing complex graph structures? I would expect that some errors are caused by LLM following the impossible to achieve instructions, while the successful cases may be merely caused by chance (where the target nodes in the incorrectly extracted graph have the same relations with the groundtruth) or by not following the ill-defined instructions.
> - **Presentation Note**:
>   - Again, the sentence on Line 247 could be clearer with a rephrase. It seems that you meant to say that with the "Graph" method, the LLM just extracts the graph, and the rest is done without the LLM but with a graph traversal executed on the extracted graph.
>   - Titles of Section D.2 and onwards: Maybe you should remove "Anthropic and Llama" since there is also GPT-5 Thinking in these tests.

---

> > ### Author Response · Authors · 2025-11-26
> >
> > Thank you for the notes! To clarify for the complex graphs, we would give the LLM all the direct edges in the graph and specify that no other direct causal edges exist. This uniquely identifies the causal DAG.
> > As such, we would tell the LLM:
> >
> > "Use this narrative N and the following direct causal relations from a predicted causal graph: A→B, A→C, B→D, C→D such that → indicates direct causation, and no other direct causal edges exist between any nodes. Did A cause B? The cause can be direct or indirect, meaning you should answer “Yes” if there is any directed path from A to B using the given edges. Output your answer with Yes/No."
> >
> > We will clarify this in the updated pdf and also clarify the presentation notes you mentioned.

---

### Official Review · Reviewer_C7XZ · 2025-11-02

**Soundness:** 2
**Presentation:** 3
**Contribution:** 3
**Rating:** 6
**Confidence:** 3

**Summary:**

This paper analyzes how large language models assess causality in events described within synthetic, semi-synthetic, and real-world narratives. The authors examine the extent to which LLMs rely on their internal (parametric) knowledge, especially when their pre-existing causal beliefs conflict with information provided in the prompt. Additionally, the study investigates how the size and complexity of narratives and causal chains impact model performance. The authors also experiment with various prompting techniques and different types of causal chains to further understand these effects.

**Strengths:**

- The paper is well written. While presenting multiple experimental settings, the authors succeed in delivering all necessary information and maintaining a coherent and accessible narrative throughout.
- The experimental setup is particularly compelling. The authors systematically vary the conditions, carefully designing experiments that isolate and test one aspect at a time, which strengthens the validity of their reasoning.

**Weaknesses:**

**[W1]** One of the main limitations of this work is that the primary analysis and results focus almost exclusively on the GPT-4o model. While the authors include results for two additional models in the Appendix, they do not provide any substantive analysis or discussion of these models. Furthermore, not all experiments are conducted on both additional models. As a result, the paper is heavily GPT-4o-centric, which limits the generalizability of the findings and misses the opportunity to provide cross-model insights (such as how model size affects reliance on parametric knowledge).

**[W2]**  The authors state that one of their motivations is to investigate the interaction between world knowledge and abstract logical reasoning, as well as the balance between reasoning and "the right amount of world knowledge". However, they do not clearly define what constitutes the "right amount" of world knowledge nor do they operationalize this concept in their experiments. It is also unclear how the causal and anti-causal relationships in their study map onto the dimensions of world knowledge and logical reasoning that they aim to explore. This claim would benefit from a clearer motivation and more precise definitions.

**[W3]** The authors claim to evaluate the models' ability to reconstruct a causal graph from a narrative. However, they do not provide quantitative results or metrics for the performance of the causal graph extraction task.

**[W4]** The paper treats anti-causality in a binary fashion, but in reality, anti-causal events can vary in the degree to which they conflict with parametric knowledge. Not all violations of common causal knowledge are equally significant. For example, the absence of a causal link between "lack of sleep" and "happiness" may be more OOD  and represent a stronger negative causality than the absence of a link between "lack of sleep" and "morning shower" where the events may simply be unrelated. In the former case, it might be desirable for the model to rely on its parametric knowledge for safety, while in the latter, reliance on the prompt would be preferable. These nuances and potential confounding factors in the data are not explored, which limits the actionability of this experiment's insights.

**Questions:**

Please refer to the weaknesses.

---

> ### Author Response · Authors · 2025-11-21
>
> Thank you for your kind remarks and taking the effort to give thorough suggestions!
>
> Thanks so much for your comments and suggestions!
>
> **W1**
>
> We have added further analysis of the results of the additional models we analyze and have added this to the pdf of the paper. We have also added an analysis of LLama 3.1 70B and GPT-5 (Thinking) High Reasoning, which allows us to survey the scope of small models to larger models: in LLama 3.1 8B, LLama 3.1 70B, Claude 3.5 Sonnet, Chat GPT 4-o and GPT-5. The new results we have added thus far in rebuttal are for the synthetic setting, but we hope to provide results for the new models for the other regimes during the remaining portion of rebuttal (pending rate limits/run time). We see that larger models like GPT-4o and Claude 3.5 Sonnet perform far worse on knowledge that conflicts with their pre-training compared to LLama models, possibly because they have been trained on so much more data than the LLama models. As such, we can say that size of the model doesn’t necessarily translate into better performance for the failure modes we identified. What does seem to translate into significantly better performance is the amount of reasoning capability the model explicitly has. GPT 5-Thinking High Reasoning which takes a very long amount of time to analyze the problem is able to achieve 85 percent accuracy in both the Anti-Causal and Reverse Regimes.
>
> **W2**
>
> Thank you for the great question. To clarify, we think of world knowledge as the parametric knowledge that the LLM has stored from memorizing facts about the world in its pre-training data. Our study of anti-causal and causal relationships plays into the interplay between world knowledge and reasoning, as the anti-causal and causal relations are learned from world knowledge and we test the LLM’s ability to reason on narratives that contradict that world knowledge the LLM has memorized. To understand what the narrative is even saying, the LLM must recall certain facts about the world from its world-knowledge. However,  the LLM must not rely solely on its world-knowledge when it comes to its memorized anti-causal and causal relations, but has to actually apply reasoning skills to its comprehension of the narrative. Thus, to be able to reason about a narrative that contradicts some amount of the LLMs preconceived biases, the LLM has to balance the “right amount of world knowledge” with its reasoning skills.
>
> **W3**
>
> The Forward vs Reverse Accuracy plot (Figure 1, left) measures the performance of all prompting methods—including the extracted graph G′ (where we ask the LLM to predict the causal graph from the narrative) labeled “Graph” and shown with a star marker—against the ground-truth graph G. In this accuracy plot, the extracted graph in the reverse direction performs far better than all other prompting methods in the reverse direction. To compute the accuracy of the extracted graph G′ , we test every possible ordered pair of variables and check whether the extracted graph correctly implies the existence and direction of the corresponding causal edge when compared to the ground truth G. This will be made more clear in the paper. We have added other metrics measuring differences between G and G’ to the paper like Graph Edit Distance as well in rebuttal.
>
> In addition, during our rebuttal, we have expanded our external analysis to be across 150 unique narratives for each of the three classes (synthetic/semi-synthetic and real-world) checked by three different reviewers (external grad students who are not authors) who rated faithfulness of G’ to G (whether they could reconstruct G from G’). In each regime, at least 97 percent of the ground truth graphs were completely correctly reconstructed.
>
>
> **W4**
>
> We agree that we have not explored the different extents of how strong the LLM bias is surrounding its prior on a causal relationship. This is an extremely interesting direction and we will address it as part of our future work.

---

### Author Response · Authors · 2025-11-30
**Summary of Contributions**

Hi! As suggested by the ICLR conference in light of recent events, we will summarize our rebuttal contributions. Firstly, we are thankful for the work of all reviewers and their broad support for the acceptance of our paper in their initial reviews, and for the work both the initial and new AC. Many reviewers asked us about what our results would look like if we expanded testing to further models, and as such during rebuttal we added Llama 70B and GPT-5 Thinking experiments to our already existing experiments for GPT-4o, LLama 8B and Claude 3.5 Sonnet. This allowed us to examine failure modes across model sizes and reasoning capabilities, and we show these results in Appendix D (Page 21). Next, reviewers asked us about our external validation process and if we had future plans to upscale it. While challenging under the time constraints, we were able to accomplish this task during rebuttal by recruiting 3 external validators to test the quality and fluency of hundreds of generated narratives under a variety of circumstances. As such, we showed that we use high quality narratives in our testing, Appendix F (Page 27).  Furthermore, we were asked clarifying questions about our methodology and provided rigorous, evidence based replies to each question. As part of this process, we computed additional metrics based on reviewer recommendations such as Graph Edit Distance for our extracted graphs (Appendix D.4, page 24). Significant changes to paper are in blue in the updated pdf. During rebuttal, we only received engagement from Reviewer npyf. During their follow up to our response, they had only one question that remained and we were able to clarify a misunderstanding. While they did not have time to respond to our clarification, we are confident we addressed their remaining concern. Furthermore, for the other 3 reviewers who did not have time to respond due to the early stoppage, we are confident we provided thorough and rigorous responses to all questions. For the new AC, we would be happy for the opportunity to answer any question that you may have. We thank you very much for taking the time to review our paper.

---

### Meta-Review · Area_Chair_ESin · 2026-01-07

**Summary:**

This paper analyzes how large language models assess causality in events described within synthetic, semi-synthetic, and real-world narratives. The authors examine the extent to which LLMs rely on their internal (parametric) knowledge, especially when their pre-existing causal beliefs conflict with information provided in the prompt. Additionally, the study investigates how the size and complexity of narratives and causal chains impact model performance.

The main concerns from reviewers include

1) Limited models selected. The primary analysis and results focus almost exclusively on the GPT-4o model.

2) Unclear definition of what constitutes the "right amount" of world knowledge nor do they operationalize this concept in their experiments.

3) Unclear prompt construction. E.g., A.1.8: Only "yes" cases are provided in the few-shot prompting text, which may bias the model.
A.1.9, C.2.3: It is impossible to represent forks/joins in complex graphs (A → (B, C) → D) with a single chain. At the end of section 4.4, it is stated that "this (complex graphs) is one area where extracting an explicit causal graph does not seem to significantly improve performance" and this defect might be the direct cause.

4) Metrics & aggregation left implicit. Accuracy = agreement with G and Consistency = agreement with G' are briefly stated but the paper lacks explicit formulas, units of aggregation, seed handling, and CI recipe in the main text.

5) Thin human validation; none beyond synthetic. Only 50/2500 synthetic narratives were audited, no inter-annotation agreement is reported, no audits were performed for semi-synthetic and real datasets, even though LLMs were used to modify them. This severely limits confidence in the labels.

**Reviewer Concerns:**

All the above concerns are addressed well, the updated paper is improved with a significant change.

**Reviewer Scores:**

Based on the situation that all concerns are addressed well, the score of this paper should at least be 6,6,6,6, which is a clear sign for acceptance.

---

### Decision · Program_Chairs · 2026-01-26

Accept (Poster)